# Clinical Impact of Supplementation with Pasteurized Donor Human Milk by High-Temperature Short-Time Method versus Holder Method in Extremely Low Birth Weight Infants: A Multicentre Randomized Controlled Trial

**DOI:** 10.3390/nu16071090

**Published:** 2024-04-08

**Authors:** Nadia Raquel García-Lara, Diana Escuder-Vieco, Marta Cabrera-Lafuente, Kristin Keller, Cristina De Diego-Poncela, Concepción Jiménez-González, Raquel Núñez-Ramos, Beatriz Flores-Antón, Esperanza Escribano-Palomino, Clara Alonso-Díaz, Sara Vázquez-Román, Noelia Ureta-Velasco, Javier De La Cruz-Bértolo, Carmen Rosa Pallás-Alonso

**Affiliations:** 1Department of Neonatology, 12 de Octubre University Hospital, 28041 Madrid, Spain; cristinadediegoponcela@hotmail.com (C.D.D.-P.); befloan@gmail.com (B.F.-A.); adclara@yahoo.es (C.A.-D.); saravazquezroman@gmail.com (S.V.-R.); noelia.ureta@gmail.com (N.U.-V.); kpallas.hdoc@gmail.com (C.R.P.-A.); 2Aladina-MGU-Regional Human Milk Bank, 12 de Octubre University Hospital, 28041 Madrid, Spain; diana.e.vieco@gmail.com (D.E.-V.); biol.kristin.keller@gmail.com (K.K.); 3Research Institute i+12, 12 de Octubre University Hospital, 28041 Madrid, Spain; jdlcruz@h12o.es; 4Department of Neonatology, La Paz University Hospital, 28046 Madrid, Spain; mcabreral@salud.madrid.org (M.C.-L.); conchi.jimenez.gonzalez@gmail.com (C.J.-G.); esperanza.escribano@salud.madrid.org (E.E.-P.); 5Institute for Health Research–IdiPaz, La Paz University Hospital, 28046 Madrid, Spain; 6Department of Pediatric Nutrition, 12 de Octubre University Hospital, 28041 Madrid, Spain; nunezramos.raquel@gmail.com; 7Clinical Research Platform IC+12, Research Institute i+12, 12 de Octubre University Hospital, 28041 Madrid, Spain

**Keywords:** donor milk, extremely low birth weight infants, Holder pasteurization, HTST pasteurization, nosocomial sepsis, late-onset sepsis, catheter-associated sepsis, central line-associated blood stream infection (CLABSI)

## Abstract

Nosocomial infections are a frequent and serious problem in extremely low birth weight (ELBW) infants. Donor human milk (DHM) is the best alternative for feeding these babies when mother’s own milk (MOM) is not available. Recently, a patented prototype of a High-Temperature Short-Time (HTST) pasteurizer adapted to a human milk bank setting showed a lesser impact on immunologic components. We designed a multicentre randomized controlled trial that investigates whether, in ELBW infants with an insufficient MOM supply, the administration of HTST pasteurized DHM reduces the incidence of confirmed catheter-associated sepsis compared to DHM pasteurized with the Holder method. From birth until 34 weeks postmenstrual age, patients included in the study received DHM, as a supplement, pasteurized by the Holder or HTST method. A total of 213 patients were randomized; 79 (HTST group) and 81 (Holder group) were included in the analysis. We found no difference in the frequency of nosocomial sepsis between the patients of the two methods—41.8% (33/79) of HTST group patients versus 45.7% (37/81) of Holder group patients, relative risk 0.91 (0.64–1.3), *p* = 0.62. In conclusion, when MOM is not available, supplementing during admission with DHM pasteurized by the HTST versus Holder method might not have an impact on the incidence of catheter-associated sepsis.

## 1. Introduction

Nosocomial infection is a frequent problem in extremely low birth weight (ELBW) infants (less than 1000 g), reaching an incidence of up to 50%. On the other hand, it is associated with increased mortality and morbidity in the short and long term [1,2,3]. Hence, in a large prospective study, worse neurodevelopment was observed in ELBW infants who had suffered an episode of nosocomial sepsis during their admission to the neonatology unit [4]. Furthermore, a considerable increase in health care costs has also been documented [2,3,5,6,7].

It has been postulated that the first 4 weeks of life of very preterm infants represent a “critical window period” for the development of their immunity. An inadequate immune response has been strongly linked to mortality and morbidity associated not only with nosocomial sepsis but also with other conditions such as necrotizing enterocolitis (NEC), bronchopulmonary dysplasia, and retinopathy of prematurity (ROP) [6,7,8,9]. This “inadequate” immune response has also been associated with poorer long-term psychomotor development [4]. In this early period of maximum vulnerability, the diet received by very preterm infants has been identified as the most influential environmental factor in the development of immunity, metabolism, and the acquisition of the gut microbiota [8,10,11]. A decrease in nosocomial sepsis has been demonstrated in preterm infants fed with their mother’s own milk (MOM) versus formula [5,6,7,8,12,13,14,15,16]. Breast milk contains bioactive factors that present antimicrobial, anti-inflammatory, and immunomodulatory capacities; they inhibit the adhesion of pathogenic germs to the gastrointestinal mucosa, promote the creation of a gastrointestinal microbiota, maintain the integrity of the gastrointestinal barrier, repair damaged areas, promote intestinal maturation and motility, and provide antioxidant protection. The protection that breast milk provides to infants against infectious diseases is mainly based on the passive transfer of large amounts of specific immunoglobulins against many different microorganisms that compensate for the deficient immunoglobulin synthesis during the infant’s first year of life. In addition, antibodies present in human milk promote long-term intestinal homeostasis by regulating intestinal microbiota and gene expression in the intestinal epithelium [8,9,10,11].

Donor human milk (DHM) is the best alternative for the feeding of very preterm infants when MOM is not available or is insufficient, as recommended by public health and scientific organizations including the World Health Organization [17], the American Academy of Pediatrics [18], and the European Society of Pediatric Gastroenterology Hepatology and Nutrition [19]. The use of DHM during the neonatal period is associated with a decreased incidence of NEC, bronchopulmonary dysplasia, and improved digestive tolerance [18,19,20]. However, no reduction in the incidence of nosocomial sepsis in preterm infants receiving DHM by the Holder method has been demonstrated [20,21,22]. To guarantee the microbiological safety of DHM, Holder pasteurization is used in most milk banks. Holder pasteurization consists of heating the milk at 62.5 °C for a long time (30 min), followed by a rapid cooling phase to below 4 °C, and therefore, it guarantees the safety of the milk, destroying all bacterial vegetative forms as well as many viruses. However, Holder pasteurization results in the complete loss of or significant reductions in certain biologically active components present in breast milk, such as immunoglobulins, cytokines, enzymes, growth factors, hormones, and oxidative stress markers [23,24,25]. Together with the elimination of the specific microbiota of breastmilk, this has been associated with the reduced benefits of feeding via DHM compared to MOM. High-Temperature Short-Time (HTST) pasteurization is an alternative thermal process that heats the milk, arranged in a thin layer, at 72 °C for 15 s in a heat exchanger. Studies have shown that HTST treatment improves the retention of biologically active compounds compared to the usual Holder pasteurization method [26,27,28,29,30]. In this context, a continuous flow HTST pasteurizer for breast milk adapted to the conditions and requirements of milk banks was designed, validated, and patented in Aladina-MGU Madrid Regional Milk Bank in cooperation with the company Sive Fluid Systems SL. (https://patents.google.com/patent/ES2600873B1/es; accessed on 1 March 2024) [31]. Since 2020, this HTST prototype has been incorporated into the clinical practice of the Milk Bank and is therefore the first HTST pasteurizer used in the clinical setting of a human milk bank. In previous research with our prototype, greater retention of immunoglobulins and anti-inflammatory cytokines after HTST treatment compared to the Holder method (the mean retention percentages for Ig A were 80% vs. 50%, respectively; for Ig G, the values were 96% vs. 60%, and for Ig M, the values were 92% vs. 20%) was observed [32]. Additionally, we found a lesser negative impact on nutritional components, growth factors, and hormones what could have an impact on growth [32,33].

Hence, we hypothesize that in ELBW infants where there is insufficient MOM in the first 28 days of life, the administration of DHM pasteurized by the HTST method versus the Holder method might reduce the incidence of nosocomial sepsis (defined as catheter-associated sepsis). The aim of this study, therefore, was to compare the incidence of nosocomial sepsis during admission in babies that received HTST-derived versus Holder pasteurized DHM.

## 2. Materials and Methods

### 2.1. Study Design and Subjects

This study was a multicentre, randomized, double-blind, parallel-group, controlled clinical trial. The study was carried out in level III neonatal units of two public hospitals (12 de Octubre University Hospital and La Paz University Hospital) in Madrid, Spain. Both units are reference centres in Spain for the care of premature infants, and they receive DHM from the Aladina-MGU Regional Human Milk Bank of the Community of Madrid, located at the 12 de Octubre Hospital. In this sense, the 12 de Octubre University Hospital was the coordinating centre.

The study was conducted in accordance with the Declaration of Helsinki, and the protocol was approved by the Ethics Committee of 12 de Octubre University Hospital Research Institute (Project identification code 18/458; date of acceptance 27th November 2018). Ethical protocol approval was ratified by the Ethics Committee of La Paz Health Research Institute.

The trial was registered in the ClinicalTrials.gov database (Identifier: NCT04424667).

We screened all preterm infants with a birth weight of less than 1000 g who were born in either of the two participating centres or transferred before 72 h of age to determine trial eligibility. Participants were recruited between July 2020 and June 2023 and randomized when one of the legal guardians consented to their child receiving DHM. Study-specific consent was proposed within 48 h of admission (“deferred consent”). In case of parental refusal to participate, the patient was subsequently excluded from the study (post-randomization exclusion).

This study was a project funded by the Instituto de Salud Carlos III (Ministry of Health, Spain), initially for 3 years (PI18/00834), 2019, 2020, and 2021. The formal authorization for the use of the HTST prototype in clinical practice (before then, it had only been used in research) was granted in early 2020. The onset of the COVID-19 pandemic further delayed the start of recruitment. For these reasons, an extension (18 months, from January 2022 to the end of June 2023) was requested and granted. Further extensions were not possible.

The following exclusion criteria before randomization were established: infants diagnosed with a chromosomopathy and/or at least one major congenital malformation, intrauterine hypoxic event (umbilical cord pH or first pH at admission < 7), and having received formula or donor milk outside the study prior to their transfer. The post-randomization exclusion criteria were as follows: legal guardians’ refusal or withdrawal, infants who started enteral feeding beyond the 7th day of life, infants that did not receive DHM in the first 28 days of life, and infants that were included in another clinical trial that modifies nutritional management during admission to the neonatal unit.

#### 2.1.1. Sample Size Calculation

We calculated that 150 children should be included in each group (total number: 300 children) to estimate a 33% reduction in the number of children suffering from sepsis with a confidence level of 95% and a power of 80%. It was planned that out of the 125 children per year admitted with the required characteristics to the two hospitals, 105 would meet the inclusion criteria, and over three years, 315 would be eligible for the study.

#### 2.1.2. Study Protocol

##### Feeding Protocol and Growth Monitoring

Before initiating the study, an enteral nutrition protocol was agreed between the two participating hospitals. Parenteral nutrition was started immediately after birth for all infants. The agreed enteral feeding protocol recommended MOM as the best feeding alternative and DHM as the second-best option. This protocol involved the early administration of maternal oropharyngeal colostrum; administration of trophic enteral feeds (<20 mL/kg/day) per orogastrical tube in the first 24 h of life (except in situations of known/suspected congenital intestinal obstruction or haemodynamic instability where it is considered that splanchnic flow may be compromised); and the progression of enteral feeds with daily increments of 20 to 25 mL/kg/day depending on digestive tolerance. Furthermore, when a volume of enteral nutrition between 80 and 120 mL/kg/day (at medical discretion) was reached, a fixed-dose multicomponent fortifier (Almirón Fortifier ^®^Nutricia, Zoetermeer, Netherlands or Prenan FM85^®^, Nestlé, Vevey, Switzerland) was added. Following the principles of adjustable fortification (weekly serum urea measurement), a protein module (Protifar^®^, Nutricia, Zoetermeer, Netherlands or Olygopeptide^®^, Nestlé, Vevey, Switzerland) was added. Moreover, fat and/or carbohydrates modules could be added to MOM or DHM at clinical discretion. If maternal milk supply was insufficient, infants received donor milk as an alternative until 34 weeks of postmenstrual age (PMA) or even further at medical discretion. Subsequently, they received formula if their mothers could no longer supply their milk. The use of medication to inhibit gastric acidity was not recommended.

No other clinical management interventions, including early-onset and nosocomial infection protocols, nor the use of probiotics were agreed between the two participating hospitals. All other intensive care and nutrition aspects were established according to usual practice.

##### Processing of Study DHM Batches

The milk bank database was adapted to allow for the correct management of batches of DHM pasteurized by the two different methods. To make up the study batches of pasteurized milk, milk was selected from donors whose previous batches were high in protein and calories. An initial mixture of donor milk from one donor or different donors was divided into 2 pools of the same volume. One of the pools was pasteurized by the Holder method, and the second pool was pasteurized by the HTST method (“paired” mixtures) (see Appendix A). For every HTST cycle, the pasteurization temperature and time were 72 °C and 15 s, respectively. The resulting aliquots of pasteurized milk were automatically labelled as A or B according to the type of pasteurization.

##### Randomization and Blinding

We assigned infants to one of the two study groups using computer-generated random block sequences stratified by centre. The milk bank staff or a neonatologist in charge (outside milk bank opening hours) submitted the clinical record number and filled out the name of each participant. Twins were randomized individually. The allocated group was recorded in the medical clinical records of each participant.

Since enteral feeding was started until 34 weeks postmenstrual age (PMA) or beyond at medical discretion, the patients included in the study who did not have sufficient MOM only received the type of pasteurized DHM assigned by the randomization sequence.

The milk bank staff or a neonatologist (outside milk bank opening hours) delivered study aliquots. During the time the study was active, a contingency fund with DHM classified as A or B was included within the neonatal unit of both centres in case study DHM was needed for an infant. For study participants, the milk bank database did not allow for the selection of DHM that had been pasteurized by any method other than that of their corresponding group. August and September 2021 recruitment was stopped due to the lack of study-specific milk stock across 2 months. The reasons for this were temporary technical problems related to the HTST pasteurization prototype.

Only the main researcher at the coordinating centre and the milk bank staff involved in the processing of DHM knew the type of pasteurization to which the letters A and B corresponded. The following groups of people were blinded to the diet group assignment: parents, milk bank, and medical staff involved in allocating DHM aliquots for the participants, all medical providers involved in making decisions regarding feeding (attending neonatologists, residents), all medical providers involved in preparing or delivering feedings to the infant (nurses), all clinicians otherwise involved in the care of the infants, and those responsible for data collection and analysis.

#### 2.1.3. Study Outcomes

The study’s main outcome measure was the comparison of the incidence of at least one bacteriological catheter-related confirmed sepsis during admission using the HTST method versus the Holder method.

NeoKissEs Nosocomial Infection Surveillance definition for nosocomial sepsis was used [34]. The German Centre for Nosocomial Infection Surveillance, which has set up a specific system for very low birth weight (VLBW, born with less than 1500 g) infants called NEO-KISS, is a leader in the implementation of the epidemiological surveillance of nosocomial bacteraemia in Europe [35]. In Spain, based on NEO-KISS, a surveillance system for nosocomial bacteraemia in VLBW infants (NeoKissEs) has been implemented, to which many neonatal units in the country adhere [34]. As a prerequisite, each patient had to carry an intravenous catheter for more than 2 days and have a line in place on the day of clinical onset or removed less than 48 h before. In case of central line replacement, the infection may be associated with the removed catheter and not with the current catheter if the removal took place less than 48 h after clinical onset. For pathogens other than Coagulase Negative Staphylococcus (CNS), to confirm infection, the isolation of bacteria from blood or cerebrospinal fluid other than CNS plus 2 or more of the following clinical signs were needed: fever, temperature instability, tachycardia or increased/new onset bradycardias, capillary refill time > 2 s, increased/new onset apnoeas, metabolic acidosis, new onset hyperglycaemia, or other signs of sepsis (skin discolouration, increased c-reactive protein (CRP) and/or interleukin, intubation, unstable general condition, apathy). For CNS sepsis, CNS must be the only pathogen isolated in the blood culture, meet at least 2 clinical criteria specified above, and additionally meet at least 1 of the following laboratory criteria: CRP > 2 mg/dl, increased Interleukin, ratio of immature/total neutrophils > 2, leucocytopenia < 5000/mcL or thrombocytopenia <1,000,000/mcL (or <100/nL).

Fungal infections were collected separately (very low incidence in both neonatal units).

As secondary outcomes, we considered other clinical events or conditions during their admission to the neonatology unit, such as mortality, confirmed catheter-associated sepsis episodes/1000 catheter days, confirmed catheter-associated sepsis and/or NEC Bell stage ≥ II, NEC that required surgery, isolated intestinal perforation, number of intolerance episodes (set of symptom–signs leading to an adjustment in feeding, clinical signs, and changes in feeding pattern on the same day were considered to be included in the same episode), O2 requirement at 36 weeks, ROP grade III or higher or need to treat it (anti-vascular endothelial growth factors-anti-VEGF- drugs, laser therapy or both), extrauterine growth restriction –EUGR- (decline from birth to 36 weeks’ PMA in weight-for-age z score more than 1 SD (Fenton curves) or weight gain velocity < 12.5 g/kg/d) [36,37,38,39,40], average stay (days), and type of feeding at discharge (exclusive breastfeeding (BF), any BF, and exclusive formula feeding).

#### 2.1.4. Data Collection

Data were collected and entered weekly into an electronic notebook database. Clinical data were extracted from the clinical records of each patient, as well as from the discharge report. All clinical events were cross-checked by the study’s main investigators in each of the centres (neonatologists). The researchers in charge of recording the data received specific training from the principal investigator. The definitions of the variables to be collected, as well as precise instructions for recording, were included in the study’s investigator’s manual.

The following data were recorded:

Birth: date of birth, sex, gestational age (weeks and days), birth weight, length and head circumference at birth (if applicable), Apgar score at 5 and 10 min, and clinical risk index for babies (CRIB) (first version).

Pregnancy: mother’s nationality, age at the start of pregnancy, pregnancy control, maternal pathology during pregnancy with possible foetal repercussions, maternal consumption of drugs or medicines, and prenatal corticoids.

Daily collected data: weight; length; head circumference; presence of catheter; number of daily doses of maternal colostrum in the first 4 days of life (immunotherapy); actual volume of MOM and DHM (mL/kg/day); feeding regime (bolus or continuous); actual intake of carbohydrates, protein, fat, and calories from parenteral nutrition; additives for MOM/DHM (multicomponent and protein fortifier, carbohydrates, and fat modules); probiotics, antibiotics, anti-acids, antivirals, vasoactive drugs, ibuprofen, or paracetamol for the treatment of patent ductus arteriosus (PDA); and postnatal corticosteroids.

Clinical events: episodes of digestive intolerance, bacteriologically confirmed catheter-associated sepsis, viriasis, fungal infection, NEC, and isolated intestinal perforation.

Discharge: weight at 36 weeks postmenstrual age (PMA) and discharge, deceased during admission, oxygen requirement at 36 weeks PMA, degree of ROP, need for ROP treatment, type of respiratory support and FiO2 36 weeks PMA, need for postnatal corticosteroids.

All patients were weighed at admission. Subsequently, in the first weeks of life, weights were measured at a variable time interval (usually every 3–7 days) depending on the clinical situation of the patient. When the patients were stable, weight was measured every 24–48 h.

The volume of MOM and DHM received from birth until 34 weeks PMA was calculated from daily records. The time cut-off point of 34 weeks postmenstrual age was chosen because beyond 34 weeks postmenstrual age, some patients were able to perform breastfeeds that did not require full tube supplementation, making it difficult to quantify the total volume of enteral nutrition consumed.

Weight z-scores and deviation were calculated based on gestational age-specific growth charts published by Fenton [36,37]. Weight gain was calculated using Patel’s formula in grams per kilogram per day [40].

### 2.2. Statistical Analyses

Patient characteristics at baseline and during admission and patient outcomes were described by the intervention group with median and interquartile ranges (IQRs) for continuous variables and with absolute frequencies and proportions for categorical variables. The statistical tests used for comparisons were the Mann–Whitney U test and Chi-square or Fisher Exact test. For comparing the proportion of children with at least one episode of sepsis between the HTST and the Holder group, we estimated the relative risk and its 95% confidence interval. We performed a predefined subgroup analysis of the main outcome by study centre. The Breslow–Day test was used for assessing homogeneity between strata. Differences with *p*-values <0.05 were considered statistically significant. A safety analysis was performed when study enrolment reached 33%. STATA (version 18), SAS (version 9.2), and R software (version 4.1.1) packages were used for statistical analyses.

## 3. Results

### 3.1. Study Population

Figure 1 shows a flow chart of the study. Overall, 290 ELBW infants were born during the recruitment period in the two centres. As 77 were not eligible for the study, 213 patients were randomized to receive the intervention. A total of 53 patients (27 from the Holder group and 26 from the HTST group) were excluded after being randomized, which implies a 25% of previously randomized patients. Of the 53 excluded patients, a total of 46 patients did not receive the intervention. A breakdown of the motives for exclusion by randomization group is presented in Figure 1.

The baseline characteristics of patients according to the type of pasteurization are included in Table 1. Both groups were homogeneous, as no significant differences were found in any baseline variable.

Additionally, the clinical variables during admission for both intervention groups are included in the Appendix A Apart from the volume of MOM consumption during the first 28 days of life and from birth to 34 weeks PMA, we did not find differences between the two groups.

For the overall cohort, the volume percentage of MOM and DHM consumed from admission to 34 weeks PMA was 82.9% for MOM versus 17.1% for DHM.

### 3.2. Main Outcome

We observed that 41.8% of the infants (33/79) included in the HTST group had at least one episode of bacteriologically confirmed catheter-related sepsis during admission, compared to 45.7% (37/81) of the infants included in the Holder group. The relative risk of suffering confirmed catheter-associated sepsis was 0.92 (0.68–1.26) for the patients supplemented with DHM pasteurized by the HTST method versus the Holder method, with a *p*-value of 0.62.

In the predefined subgroup analysis by study centre (see Appendix A), the relative risk was 0.73 (0.51–1.03) in centre 1 and 1.23 (0.73–2.11) in centre 2 (difference between strata *p*-value = 0.09).

In addition, baseline and clinical variables during admission per centre are included in the Appendix A

### 3.3. Secondary Outcomes

Table 2 shows the results of the secondary outcomes according to pasteurization group. There were no significant differences between the two pasteurization groups and other clinical events and conditions.

We found only five fungal infections in five different patients, all of them of the genus Candida; one of these patients belonged to the HTST group, and four patients belonged to the Holder group.

## 4. Discussion

This is the first study using a clinical trial methodology that evaluates the clinical impact of supplementing MOM with DHM pasteurized by two different methods (HTST vs. Holder) in ELBW infants during admission. No significant differences were observed in the number of children with at least one episode of bacteriological confirmed catheter-related sepsis according to pasteurization method. The frequencies found were 42% for the HTST method and 46% for the Holder method.

In our study, a continuous flow HTST pasteurizer designed, validated, and patented in Aladina-MGU Madrid Regional Milk Bank in cooperation with the company Sive Fluid Systems SL was used (https://patents.google.com/patent/ES2600873B1/es; accessed on 1 March 2024). For every HTST cycle, the pasteurization temperature and time were 72 °C and 15 s, respectively. The technique used for a specific HTST prototype (continuous flow or batches), as well as the time and temperature parameters (as has also been described for other processing techniques for DHM), impacts differently on the bioactive components of DHM [24]. Therefore, it is very important to specify, both in laboratory and clinical studies, the particular HTST prototype, as well as specific parameters such as the temperature and time of pasteurization used in order to correctly evaluate the results.

Research that focuses on new processing methods for DHM which is able to reach a better balance between quality and safety is a priority for milk banks worldwide [23,24,25,26,27,28,29,30]. As previously published [31,32,33], our prototype showed a less negative impact regarding the retention of key components such as immunoglobulins, cytokines, enzymes, and hormones. However, to the best of our knowledge, this is the first study that focused on the clinical impact of a new processing technique for DHM in ELBW infants.

This clinical trial reproduced the standard practice in most third-level neonatal units in Europe, including the high percentages of prenatal corticosteroids, high percentage of MOM feedings during the first weeks of admission, supplementation with DHM until at least 32–34 weeks of corrected age, adjustable fortification (cow milk fortifiers), and close growth monitoring [22].

Consent was given on a deferred basis by one of the investigators. We aimed to not delay the initiation of enteral feeding and to avoid the information overload that parents of these large preterm infants experience in the first hours after birth. The refusal rate was low, as only 3.8% (8/213) of the legal guardians refused to participate.

Our study hypothesis was not confirmed. For ELBW infants who did not have enough MOM in the first 28 days of life, the administration of DHM pasteurized by the HTST method versus the Holder method did not reduce the incidence of nosocomial sepsis.

There is great heterogeneity in the definition of nosocomial sepsis in the neonatal population. We used the NeoKissEs surveillance program [34,35] definition for nosocomial sepsis, which has blood culture positivity as the main criterion for diagnosis but additionally requires the presence of clinical signs of infection and adjunctive laboratory data. Also, it establishes specific conditions for CNS infections. This precise definition makes it easier to compare results with other studies.

There are multiple factors influencing the incidence of nosocomial sepsis, as has been stressed by recent reviews [2,3,41,42]. In this way, strategies for decreasing this pathology include interventions at multiple levels [42,43,44,45]. DHM volume received during the first 28 days and during admission was not different between the groups and was low for both groups compared to MOM volume. On the other hand, the HTST group received a significantly higher volume of MOM during the first 28 days of life and admission (a protective factor against nosocomial infection), but this did not translate into a lower incidence of sepsis in the HTST group. The observed feeding pattern in our population (patients received high volumes of MOM and small volumes of DHM during admission) makes it difficult to assess the clinical impact of administering immunologically “richer” DHM. In a previously published clinical trial, enrolled patients who did not receive sufficient MOM were supplemented in the first 10 days of life with DHM versus formula to assess the impact on the incidence of nosocomial sepsis. They did not find differences in the incidence of nosocomial sepsis between the groups, and similar to our study, 90% of feedings during the time of intervention were MOM feedings [22]. Another consideration is that although DHM pasteurized with the HTST prototype has a better immunomodulatory and anti-infective protein profile, this treatment eliminates all presence of vegetative bacteria, including the native microbial flora of breast milk, with its beneficial effects on the bacterial intestinal colonization of preterm infants. This fact could justify the absence of a protective effect.

This clinical trial was rather pragmatic. Only the clinical protocol related to enteral nutrition was agreed upon; the protocols for early-onset and nosocomial sepsis were different for each centre. This may account for the differences observed. In fact, some baseline and admission variables showed a significant difference between the two centres (Appendix A). Although there were no differences in median gestational age and birth weight, centre 1 had a higher percentage of low birth weight infants (probably due to it having a higher number of infants with intrauterine growth retardation). In terms of baseline variables, there were no differences in catheter indwelling or parenteral nutrition. Considering clinical variables during admission, it can be observed that, in centre 1, the study participants received 9 days less antibiotic treatment during admission and 7 days less in the first 28 days of life and received a higher proportion of probiotic treatment. In addition, they received anti-acid treatment much less frequently. Regarding the DHM and MOM received, the patients in both centres received a similar volume of MOM during the first 28 days of life and admission but a different volume of DHM during the first 28 days and admission. Although the DHM volume received by patients in both centres was remarkably low, patients in centre 2 received a significantly smaller volume of DHM than patients in centre 1. In summary, centre 1, with patients with a higher incidence of low birth weight infants (which correlates with intrauterine growth retardation, a reported risk factor for nosocomial sepsis), less frequently uses antibiotics and anti-acid medication and more frequently uses probiotics. These three variables have been associated with a lower incidence of nosocomial sepsis [3,41,42,43,44,45,46,47,48,49,50,51] and may at least partly explain the differences found between the centres. On the other hand, the very low volume of DHM received by patients in centre 2 makes it difficult to assess the impact of the type of pasteurization of DHM on the incidence of nosocomial infection in this centre.

Focusing on secondary outcomes (all related to admission period), according to pasteurization group, we found no differences in the rate of confirmed catheter-associated sepsis episodes per 1000 catheter days or the frequency of the combined outcome of catheter-associated sepsis and NEC, NEC that required surgery, isolated spontaneous perforation, need for oxygen at 36 PMA, ROP grade III or greater, ROP that required treatment, and type of feeding at discharge. Additionally, the data for hospital stay, number of intolerance episodes per patient, and growth parameters (regarding weight) were similar between both groups. The rates of EUGR were high, similar to those reported in previous studies [38,39]. Despite the more favourable immunological profile of DHM pasteurized with the HTST prototype, no differences were found in conditions where an inadequate immune response was implicated, such as NEC, broncopulmonary dysplasia, or ROP. The lesser impact on hormones, enzymes, and nutritional parameters demonstrated by the HTST prototype was not reflected in a decrease in episodes of digestive tolerance or growth in our study. The lack of an effect could be explained by the low incidence of some conditions, such as NEC or isolated intestinal perforation (the sample size was calculated to evaluate nosocomial sepsis, a much more prevalent pathology); the low volume of DHM consumed during admission; or the multifactorial nature of many of the conditions included as secondary outcomes.

One limitation of our study is that we did not reach the planned sample size and statistical power. Fewer patients were born in the two centres than expected (considering previous trend from both centres). The present study was carried out between 2020 and 2023. In both centres, as reflected worldwide, the SARS-CoV-2 pandemic led to a decrease in preterm baby births during the lockdown period [52,53]. In addition, a higher proportion of patients than expected were not considered initially eligible, and another significant proportion of infants were excluded after randomization. This is justified by the extreme vulnerability of the study population (high percentage of early-life death, clinical severity in the first weeks of life preventing the initiation of first week of life feeding). Considering the sample size calculation, we considered a 33% reduction in nosocomial sepsis by HTST pasteurization, based on our previous research [32]. Greater retention of immunoglobulins after HTST treatment compared to the Holder method was shown (the mean retention percentages for Ig A were 80% vs. 50%, respectively; for Ig G, the values were 96% vs. 60%, and for Ig M, the values were 92% vs. 20%). However, it is important to highlight that for sample size calculation, we assumed a higher DHM volume consumed during admission than that found in our study.

Another limitation of our study is that we did not include long-term outcomes. Finally, regarding growth, body composition measurements are considered of great interest for better assessing optimal growth in ELBW infants. Body composition (vector electrical bio-impedance) parameters were only partially collected, so this factor was not analysed.

## 5. Conclusions

This study did not find differences in the frequency of catheter-associated sepsis in ELBW infants that were supplemented (when necessary) with DHM pasteurized by the HTST method versus Holder method during admission. Additionally, no differences were observed regarding other relevant outcomes, such as mortality, surgical NEC, isolated bowel perforation, BPD, ROP, hospital stay, type of feeding at discharge, and EUGR. The calculated sample size was not reached. Additionally, the large amount of MOM and the small amount of DHM consumed by the participants, as well as the involvement of multiple factors in the development of nosocomial sepsis, could have contributed to the lack of effect of the intervention.

To our knowledge, this study is the first randomized trial to explore the clinical impact of supplementing DHM pasteurized by a novel method that allows for a higher quality product that is nearer to MOM. Mortality, main morbidities, and growth during admission were evaluated. Further research is required to establish the long-term effects of supplementing with DHM processed with this prototype. In the future, for new patented prototypes for processing DHM, it is highly recommended to evaluate their clinical impact following a randomized clinical trial methodology.

## Figures and Tables

**Figure 1 nutrients-16-01090-f001:**
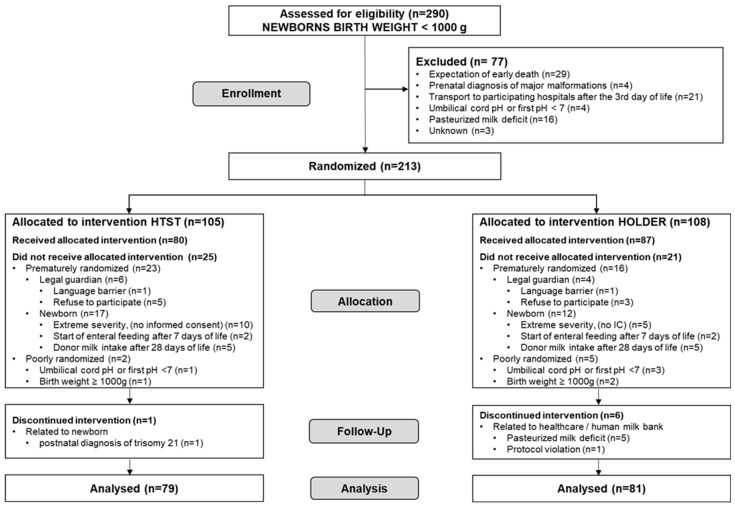
Flow chart of the study.

**Table 1 nutrients-16-01090-t001:** Baseline characteristics according to pasteurization group.

	HTST Group (N = 79)	Holder Group (N = 81)	*p*-Values
	N (%)	N (%)	
Sex (Male)	35 (43.2)	34 (43)	0.98
Multiple pregnancy (≥2)	26 (32.9)	27 (33.3)	0.95
Pregnancy (well-controlled)	72 (91.2)	74 (91.4)	0.96
Country of origin Spain	47 (61.8)	49 (62.8)	0.90
Antenatal corticosteroids (≥1 full course)	62 (78.5)	65 (80.3)	0.78
Low weight for gestational age (*p* < 10)	24 (30)	22 (27)	0.65
	Median (IQR)	Median (IQR)	
Gestational age (weeks)	26 (25; 27)	27 (25; 28)	0.93
Birth weight (grams)	820 (650; 917)	817 (660; 900)	0.95
Fenton z score for birth weight	−0.36 (−1.4; +0.2)	−0.6 (−1.3; +0.1)	0.75
Apgar score 5 min	8 (6; 9)	8 (6; 9)	0.62
CRIB I score	6 (3; 9)	6 (2; 9)	0.93

Qualitative variables are presented as the absolute and relative frequencies (%). Quantitative variables are presented as medians (interquartile range; 25th, 75th percentiles) IQR: interquartile range; CRIB: clinical risk index for babies, first version.

**Table 2 nutrients-16-01090-t002:** Secondary outcomes according to pasteurization group.

	HTST Group (N = 79)	Holder Group (N = 81)	*p* Value
No. episodes of confirmed catheter-associated sepsis/1000 catheter days	17.5	18.7	0.78
	N (%)	N (%)	
Mortality	4 (5.1)	4 (4.9)	0.97
NEC requiring surgery	5 (6.3)	2 (2.5)	0.23
NEC Bell grade ≥II or confirmed sepsis	36 (45.6)	39 (48.2)	0.74
Spontaneous intestinal perforation	0 (0)	2 (2.5)	0.16
Oxygen requirement at 36 weeks PMA age	21 (26)	22 (28.2)	0.82
ROP of prematurity ≥III grade	10 (13.5)	8 (10.3)	0.53
ROP requiring treatment	10 (14.5)	13(18)	0.54
Type of feeding at discharge			0.11
Exclusively breastfeeding	35 (44)	21 (26)	
Any amount of BF	16 (20)	22 (27)	
Exclusive formula	25 (32)	34 (42)	
Moderate EUGR birth—36 weeks PMA age (weight gain < 12.5 g/k/g/d)	31 (39)	36 (44)	0.50
Moderate EUGR birth—36 weeks PMA age (weight for age Fenton’s delta z-score < −1)	46 (58)	49 (60)	0.77
	Median (IQR)	Median (IQR)	
Nº of episodes of digestive intolerance/patient	1 (0; 3)	1 (0; 3)	0.79
Hospital stay (days)	85 (74; 105)	79 (69; 107)	0.95
Weight gain birth—36 wk PMA age (g/kg/day)	12.8 (11.6; 14.8)	12.7 (11.2; 14.1)	0.33
Fenton’s delta zscore for weight from birth to 36 weeks PMA age	−1.3 (−1.7; −0.8)	−1.4 (−1.9; −0.8)	0.27

Qualitative outcomes are expressed as the absolute and relative frequencies (%). Quantitative outcomes are presented as median (interquartile range). NEC: necrotizing enterocolitis; ROP: retinopathy of prematurity; PMA: postmenstrual age; EUGR: extrauterine growth restriction.

## Data Availability

The data presented in this study are available on request from the corresponding author. The data are not publicly available due to privacy issues.

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
