# Peer review of "Clinical Impact of Supplementation with Pasteurized Donor Human Milk by High-Temperature Short-Time Method versus Holder Method in Extremely Low Birth Weight Infants: A Multicentre Randomized Controlled Trial"

_nutrients, 2024, doi:10.3390/nu16071090_

Round 1

Reviewer 1 Report

Comments and Suggestions for Authors

Manuscript Nutrients-2939317

The manuscript entitled “Clinical impact of supplementation with pasteurized donor human milk by High Temperature Short Time versus Holder method in extremely low birth weight infants. A multicenter randomized controlled trial.” by García-Lara, NR. et al. provides compelling insights into the clinical impact of supplementing pasteurized donor human milk using High Temperature Short Time versus Holder method in extremely low birth weight infants, offering valuable implications for the clinical practice. The manuscript is overall well-written and organized. I found the content to be engaging and well-researched and thus suitable for publication. However, I have a few comments and/or suggestions for improvement:

·         Lines 317-324. The values provided in the text regarding the relative risk to suffer a confirmed catheter-associated sepsis do not align with those depicted in Figure S2. This error must be amended as it accounts for the main outcome of the study. For Figure S2, indicate “Center 1” and “Center 2” for clarity and that the relative risk refers to suffering a confirmed catheter-associated sepsis (at least at the figure caption).

·         Lines 431-439. Why didn't the authors consider extending the recruitment if the initially established sample size was not reached?

·         Since no agreement in the use of probiotics between the centers was reached (lines 161-164). I would find very useful to provide the percentage of preterm infants that received probiotics among the study. Do the authors think that a potential relationship with lower DHM volumes and higher rates of OMM could be ascribed to the use of probiotics as it could be inferred from Table S1 and S2? Please, discuss.

·         Lines 363-364: the hypothesis of the researchers hasn’t been stated, although the reader assumes that a lower incidence of nosocomial sepsis in HTST supplementation was expected by the authors.

·         Lines 406-408. The authors state that IUGR is a reported risk for nosocomial sepsis and that a higher incidence of this condition in center 2 might be responsible of the lower DHM volumes compared with center 1. However, incidence of IUGR is not reported, but percentage of low weight for gestational age (p < 10) (Table S2). This should be clarified.

·         Apgar score at 5 and 10 minutes is included among the recorded data (line 249), but only Apgar score at 5 minutes is included in Table 1 and Table S2.

·         Table 2. Spontaneous intestinal perforation instead of Isolated Spontaneous perforation?

·         Table S1

o   IQR in Days of parenteral nutrition for the HTST group is not presented as an interval as for the other variables.

o   Include the word “days” in variable “MOM volume first 28 days of life (mL/kg/day)”

o   Include the word “for” in variable “Received paracetamol for PDA (admission)”

o   Express median and IQR values as well as percentage consistently for all variables.

·         Table S1, S2 and S3.

o   I suggest to indicate in bold p-values < 0.05.

·         Line 259: anti-acids and antacids. Duplicity?

·         Line 320: past tense would be more appropriate.

·         Line 389-390: This sentence is incomplete.

·         I recommend revising the use of abbreviations to ensure they are used appropriately and consistently throughout the text (e.g. PMA, wk, IC, BF). Consider defining each abbreviation upon its first use and maintaining consistency thereafter.

·         The manuscript could benefit from a thorough review of punctuation usage. Consistency in the expression for decimal positions in numbers is also encourage (Tables 1, 2, S1, S2).

Reviewer 2 Report

Comments and Suggestions for Authors
    • This article evaluates the clinical impact of administration of donor human milk pasteurized with two differents methods(holder versus HTST) in supplementation of own mother milk, It is the first randomized, multicenter study evaluation clinical impact
    • The ethics statement is respected
    • The  manuscript is clear, relevant for the field and presented in a well-structured manner.

    ·         The randomized group are comparable and blindness respected. The experimental design is appropriate but he secondary outcomes are more difficult to interpret due to low occurrence

    • The cited reference are relevant and appropriate but the reference 54 et 55 are missing (cf line 435 of the manuscript). Self-citations are appropriate
    • the article Highlight the lack of number of patients initially calculated in the design of the study this weakness has been well described in the discussion. The estimated difference of 33% of reduction of number of patient suffering from sepsis was probably overestimated as the difference between HTST and holder was about 30% of difference in Ig and the proportion of MOM under estimated as the population received only 17.1% of DHM versus 82.9% of MOM
    • It could be interesting to have a look to the result in the group of infant for who proportion of  MOM  is poor as important presence of MOM  may mask the effects of the type of pasteurization of the milk

    ·         The method of holder pasteurization and of HTST (temperature, time and continuous flow or batches)) must be described and the devices used cited with reference of the manufacturer. Indeed, as depending of the devices and the quality of control of the device, the preservation of the immunological properties can be different. This is particularly important in order to be reproducible and compared in others studies.

    ·         The sentence line 389 is not finished and need to be corrected

    ·         The supplementary figure S2 is not in adequation with the text (line 322): in the text relative risk was 0.47 (0.19-1.13) in center 1 and 1.47 (0.5-3.95) in center 2; in the figure RR is 0.73 (0.51-1.03) in center 1 and 1.23 (0.72-2.11) in center 2. The interpretation is different depending of the values. The values must be corrected or in the text or in the supplementary material

    ·         The conclusion is consistent with the results but it would be interesting to add “the large” amount of MOM in addition to the  small amount of DHM to justify the lack of effect of intervention

Reviewer 3 Report

Comments and Suggestions for Authors

This multi-center RCT to investigate whether HTST treatment of donor breast milk could reduce sepsis in premature infants was designed to have immediate beneficial translation.  The hypothesis was not supported by the data.  The power calculations were assuming a 33% reduction in sepsis with HTST.  What was that assumption based on?

In the flow diagram there was a high number who did not receive the allocation due to premature randomization.  Why did this happen?
